# Uncovering the Depletion Patterns of Inland Water Bodies via Remote Sensing, Data Mining, and Statistical Analysis

Babak Zolghadr-Asli [1,2,*], Mojtaba Naghdyzadegan Jahromi [3,4], Xi Wan [2], Maedeh Enayati [5],
Maryam Naghdizadegan Jahromi [6,7], Mohsen Tahmasebi Nasab [8], John P. Tiefenbacher [9]
and Hamid Reza Pourghasemi [10]

1   Sustainable Minerals Institute, The University of Queensland, Brisbane 4072, Australia
2   The Centre for Water Systems, University of Exeter, Exeter EX4 4QD, UK; xw355@exeter.ac.uk
3   Department of Water Engineering, College of Agriculture, Shiraz University, Shiraz 7144165186, Iran;
    naghdyzadegan@gmail.com or mojtaba.naghdyzadeganjahromi@ugent.be
4   Hydro-Climate Extremes Lab (HCEL), Ghent University, 9000 Ghent, Belgium
5   Department of Irrigation & Reclamation Engineering, Faculty of Agricultural Engineering & Technology,
    College of Agriculture & Natural Resources, University of Tehran, Karaj 3158777871, Iran;
    maedeh.enayati@ut.ac.ir
6   Department of Remote Sensing and GIS, Faculty of Geography, University of Tehran, Tehran 1417853933, Iran;
    naghdizadegan.m@ut.ac.ir or maryam.naghdizadegan.1@ulaval.ca
7   Center for Research in Geospatial Information and Intelligence, Department of Geomatics Sciences,
    Laval University, Quebec, QC G1V 0A6, Canada
8   Department of Civil Engineering, University of St. Thomas, 2115 Summit Avenue, St. Paul, MN 55105, USA;
    mohsen.nasab@stthomas.edu
9   Department of Geography, Texas State University, San Marcos, TX 78666, USA; tief@txstate.edu
10  Department of Natural Resources and Environmental Engineering, College of Agriculture, Shiraz University,
    Shiraz 1352467891, Iran; hamidreza.pourghasemi@yahoo.com
*   Correspondence: bz267@exeter.ac.uk or b.zolghadrasli@uq.net.au

**Abstract:** Addressing the issue of shrinking saline lakes around the globe has turned into one of the
most pressing issues for sustainable water resource management. While it has been established that
natural climate variability, human interference, climate change, or a combination of these factors can
lead to the depletion of saline lakes, it is crucial to investigate each case and diagnose the potential
causes of this devastating phenomenon. On that note, this study aims to promote a comprehensive
analytical framework that can reveal any significant depletion patterns in lakes while analyzing the
potential reasons behind these observed changes. The methodology used in this study is based
on statistical analysis, data mining techniques, and remote sensing-based datasets. To achieve the
objective of this study, Maharlou Lake has been selected to demonstrate the application of the
proposed framework. The results revealed two types of depletion patterns in the lake's surface area:
a sharp breaking point in 2007/2008 and a gradual negative trend, which was more pronounced in
dry seasons and less prominent in wet seasons. Furthermore, the analysis of hydro-climatic variables
has indicated the presence of abrupt and gradual changes in these variables' time series, which
could be interpreted as a signal that climate change and anthropogenic drought are changing the
basin's status quo. Lastly, analyzing the statistically significant correlation between hydro-climatic
variables and the lake's surface area showed the potential connection between the observed changing
patterns. The results obtained from data mining models suggest that Maharlou Lake has undergone a
morphological transformation and is currently adopting these new conditions. If preventive measures
are not taken to revive Maharlou Lake, the tipping point might have been reached, and reviving the
lake could be improbable, if not impossible.

**Keywords:** climate change; remote sensing; time series analysis; data mining; artificial neural
network; environmental monitoring; shrinking lake

## 1. Introduction

A school of thought has gained traction among scholars recently that argues that many of the changes observed in water bodies in the recent past are at least partially attributable to human activities [1]. One of the most notable examples of such impacts is the change in inland water bodies, particularly saline lakes. A total of 44% of all lakes, volumetrically speaking, consist of large saline lakes [2]. However, many saline lakes worldwide are shrinking at alarming rates [3,4]. In addition to endangering habitats and ecosystems, such phenomena could potentially lead to human health hazards as these lakes continue to deteriorate [5].

In certain instances, human activities have been identified as the direct cause of such occurrences. Lakes Alemaya and Hora-Kilole, Ethiopia [6]; the Dead Sea, West Bank [7]; Lake Ebinur, China [8]; and Lake Corangamite, Australia [9] are merely a few examples of such cases. The Aral Sea and Owens Lake are two classic examples of the desiccation of salt lakes [5]. In other cases, climate change has been identified as the underlying cause of the issue [10]. Natural variation in hydro-climatic variables is another well-studied reason behind shrinking patterns in lakes [11,12]. More often, however, a combination of the previously mentioned factors is the reason behind the deterioration of an inland water body [13–16]. It is worth noting, however, that detection, let alone understanding the cause of such deterioration, is a challenging task [17].

The computational challenges of identifying such changes notwithstanding, detecting and, in turn, understanding such depletion patterns in a timely fashion are quite crucial from the water resources management perspective. It is worth noting that often the uncertainty associated with these patterns is one of the main notable challenges to understanding and, in turn, unraveling these depletion patterns. That said, the idea is that, should unnatural causes be detected during the investigation, one can attempt to mitigate or halt these adverse impacts. On that note, implementing remote sensing (RS)-based data to monitor and unveil the depletion patterns of inland water bodies has proven to be an effective practice e.g., [18–22]. It should be noted, however, that while RS can help extract vital raw data, additional data processing schemes are required to detect such deterioration patterns. To that end, some scholars have used statistical frameworks to discern the causes of shrinking surface area e.g., [17,23]. Although limited examples are available for this, machine-learning-oriented methods are another viable alternative to interpreting hydro-climatic data to reveal these changing patterns in inland water bodies e.g., [24]. Often, these studies rely solely on classification or clustering methods to unveil such patterns. An interesting angle is incorporating regression data mining methods such as artificial neural networks (ANN) to monitor these patterns from a more numeric-oriented perspective. The added benefit of such an approach would be that the results obtained from the data mining models would be complementary to the statistical-based analysis. On that note, this study utilizes both statistical and data mining-oriented analysis to evaluate hydro-climatic and RS-based data as a potential means to unveil hidden depletion patterns in inland water bodies. Lake Maharlou, Iran, is a case study to demonstrate this idea. Maharlou, once considered a permanent inland water body in southwestern Iran, may have reached a critical condition, and seasonal drying of the lake has become a recurring phenomenon [25], which makes it an ideal case to explore the potential behind this framework.

## 2. Study Area

The Maharlou Lake basin (Figure 1), with a catchment area of 4720 km$^2$, is located in southwestern Iran in the area circumscribed by a rectangle at 29.32°–29.55° N and 52.69°–52.90° E. While the basin normally (based on data from 1987 to 2016) experiences mild variations in rainfall and temperature (Figure 2), the southern part of the watershed receives an average of approximately 250 mm of precipitation annually, and the northern and central parts of the basin receive as much as 480 mm. The basin's average annual rainfall is approximately 390 mm. Average annual temperatures range from 18 to 19 °C, with a regional average of 19 °C [26]. Due to the region's semi-arid climate, the stream

network of the region is ephemeral; in most cases, streams and other water bodies only appear during wet seasons [26].

Maharlou Lake is a local landmark, a recreational site for locals, and a considerable environmental and economic asset for the basin. At an elevation of 1460 m.a.s.l., the lake lies in a vast endorheic basin surrounded by mountains that reach 2800 m.a.s.l. It is worth noting that the lake is mostly recharged via surface water, with no recharge points from the region's groundwater system. Despite observed historical patterns, the lake's water volume was, until recently, relatively consistent in both the wet (winter and spring) and dry (summer) seasons. Since the early 2000s, however, the lake's reservoir has seemingly experienced more pronounced fluctuations. The lake's depth varies seasonally from 0 m in the dry season to about 3.5 m in the wet seasons, and the lake's ponded area spreads to 275 km² [25]. This begs the question: are the recently observed fluctuating patterns in line with the historic hydro-climatic behavior of the lake? If not, at what point did such changes start revealing themselves?

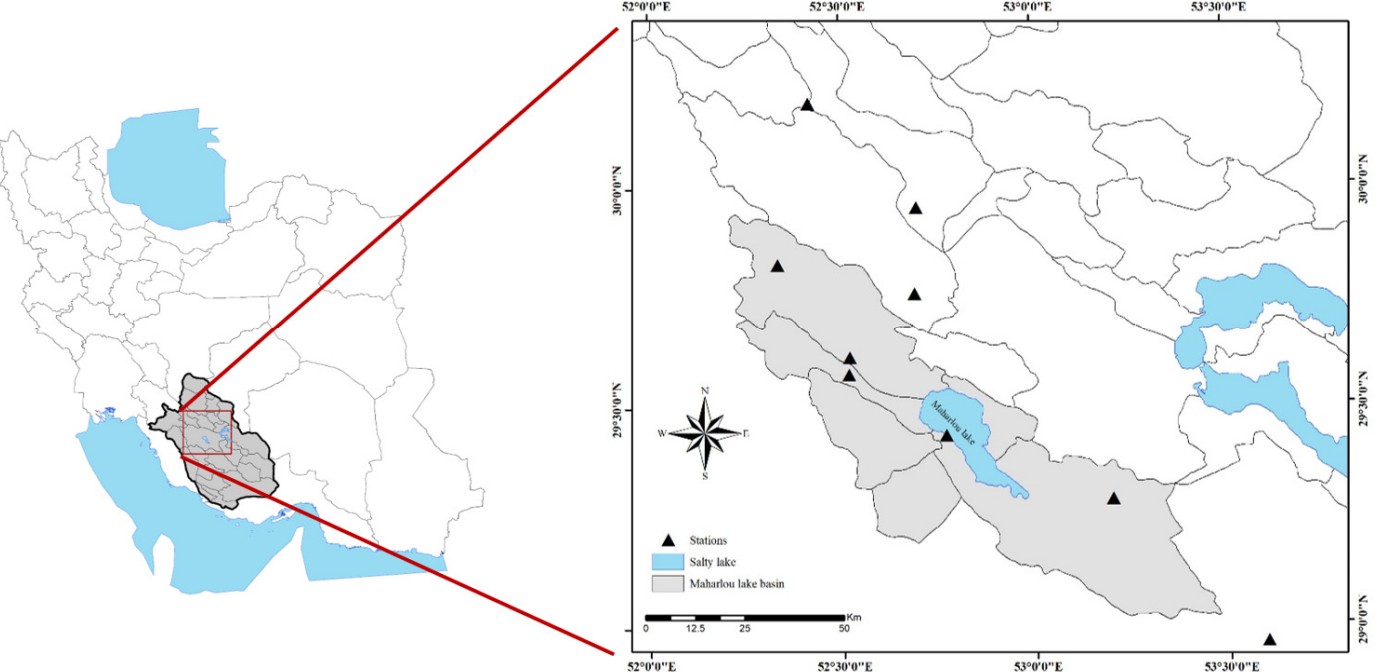

**Figure 1.** The location of the study area and selected weather stations.

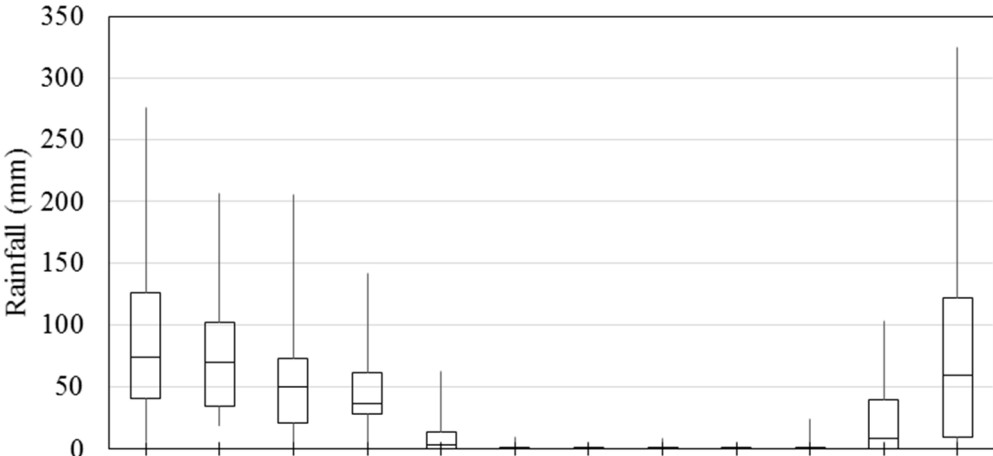

**Figure 2.** *Cont.*

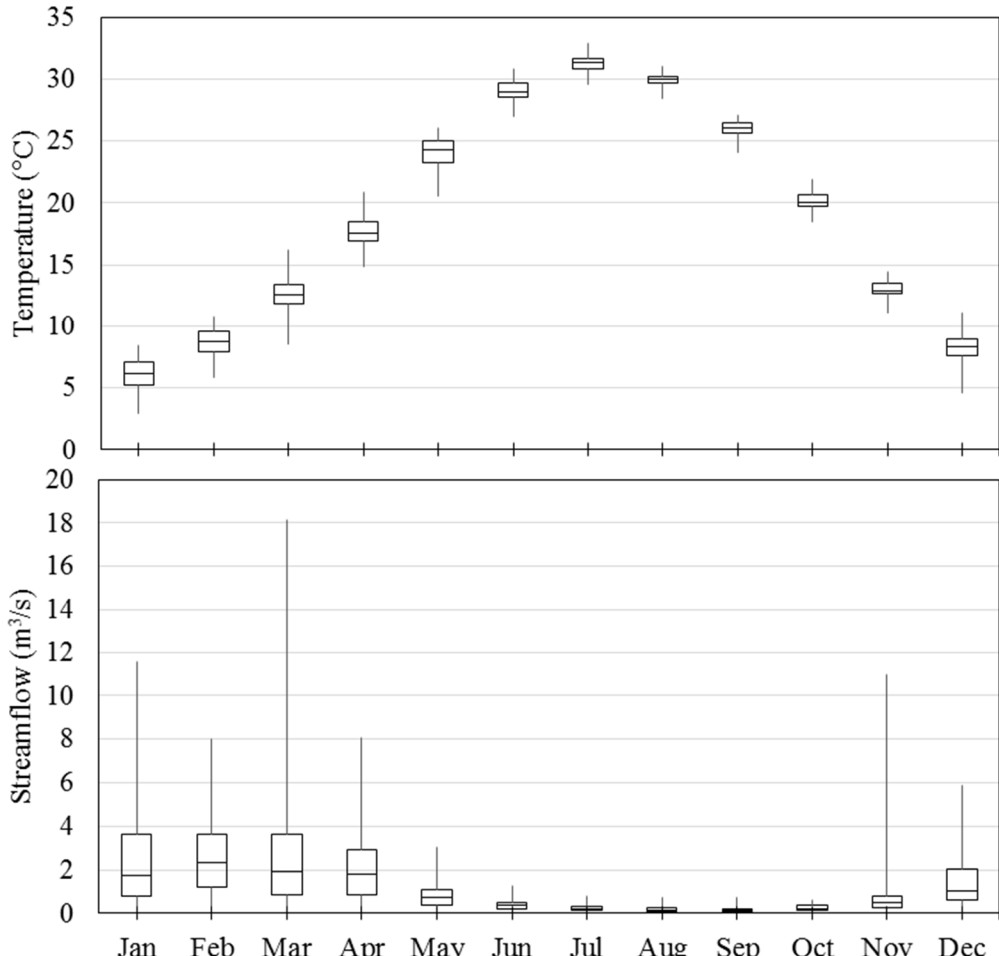

**Figure 2.** Average monthly variation of rainfall, temperature, and streamflow in the Maharlou Lake watershed from 1987 to 2016.

## 3. Methodology

This study tests a systematic framework to investigate changing patterns in the surface area of an inland water body. In the first stage, a remote sensing-based algorithm is used to delineate the lake's surface extent over a monthly time step from 1992 to 2017. Unusual changes in the lake's surface area and three additional hydro-climatic variables (i.e., precipitation, temperature, and streamflow) are investigated through conventional statistical-based time series analysis. To that end, the Pettitt test [27] is used to examine the data for statistically significant shifts in trends, and the combination of the Mann–Kendall test [28,29] and Sen's slope estimator [30] is employed to measure these possible trends quantitatively. The Spearman's test [31] is then used to conduct correlation tests between the hydro-climatic variables and the lake's surface area measurements. As the final leg of this framework, a data mining-oriented model (i.e., ANN) is used to investigate the hidden patterns in the lake's surface area data set. This framework would help further investigate the presence of any irregular behavior in the data.

### 3.1. Detection of Water Bodies

RS is a very efficient tool for monitoring spatio-temporal variations of the extent of surface water at a large scale [32,33]. Accordingly, optical sensors such as the Moderate Resolution Imaging Spectroradiometer (MODIS), Landsat, HuanJing satellite constellation-1 (HJ-1), GaoFen-1 (GF-1), Synthetic Aperture Radar (SAR), Phased Array type L-band SAR (PALSAR), and Sentinel-1A have been used for water body change detection [34–37]. Compared to images produced by other satellites, Landsat images have proven to be more

efficient for distinguishing water pixels from non-water pixels, primarily due to their long history of data acquisition and their resolution [38,39]. So, Landsat imagery has been used in this research. It should be noted that the images from the said dataset were free of clouds, and as such, no additional image pre-processing was required.

Water bodies absorb in the near-infrared (NIR) and shortwave-infrared (SWIR) and reflect in the green and blue portions of the visible spectrum, so water and vegetation-covered surfaces can be distinguished from each other. Accordingly, various indices based on these main spectra for water-body extraction from imagery have been proposed. These indices include the normalized difference water index (NDWI) [40], the modified normalized difference water index (MNDWI) [41], the high-resolution water index [42], and the automatic water extraction index [43]. The MNDWI segregates and detects water bodies and can be mathematically expressed as follows [44]:

$$\text{MNDWI} = (\Gamma - M)/(\Gamma + M) \tag{1}$$

where $\Gamma$ is the green band and $M$ is the middle infrared band. To minimize the effects of vegetation on water-body mapping, vegetation indices can be applied. Pixels with water signals greater than vegetation signals would be classified as water, while the rest would be classified as non-water.

### 3.2. Pettit Test

The Pettit test is a nonparametric statistical test that is widely used to identify monotonic jump points in a hydro-climatic data time series [17,45–47]. The null hypothesis ($H_0$) is that there is no abrupt change in the given time series. However, an alternative hypothesis (HA) is a statistically significant monotonic change-point in the time series. For a time series of continuous data $x_i$, the test statistic $U_{t,N}$ is calculated at the $t$th time step [27]:

$$U_{t,N} = \sum_{i=1}^{t} \sum_{j=t+1}^{N} \text{sgn}(x_i - x_j) \; \forall i \tag{2}$$

where $N$ = the sample size; $i$ and $j$ = the $i$th and $j$th time step (e.g., year; season; month), respectively; and

$$\text{sgn}(x_i - x_j) = \begin{cases} 1, & x_i > x_j \\ 0, & x_i = x_j \\ -1, & x_i < x_j \end{cases} \tag{3}$$

The location of the change-point ($K_N$) is:

$$K_N = max|U_{t,N}| \tag{4}$$

The significance probability ($\alpha$) of the change-point is approximately [17,45]:

$$\alpha \cong 2 \exp\left( \frac{-6K_N^2}{N^3 + N^2} \right) \tag{5}$$

Low $\alpha$-values indicate that the null hypothesis should be rejected; a significant change-point divides the time series into pre- and post-change segments.

### 3.3. Mann–Kendall Test

The Mann–Kendall test [28,29] is a ranking-based, nonparametric method widely used to investigate statistically significant monotonic trend segments in hydro-climatic time series datasets [48–50]. The null hypothesis is that the data are identically distributed and come from a population with independent realizations. The alternative hypothesis is that

the time series set contains a monotonic trend component. The Mann–Kendall test statistic ($S_{MK}$) is defined [28,29]:

$$S_{MK} = \sum_{j}^{N-1} \sum_{i=j+1}^{N} (x_i - x_j);$$

(6)

when the number of data points is equal to or greater than 8 ($N \geq 8$), $S_{MK}$ is assumed to be asymptotically normal with a variance ($\sigma^2_{S_{MK}}$) equal to [17]:

$$\sigma^2_{S_{MK}} = \frac{N(N-1)(2N+5)}{18}.$$

(7)

The Mann–Kendall test statistic ($Z_{MK}$) is calculated as follows [28,29]:

$$Z_{MK} = \begin{cases} \frac{S_{MK}-1}{\sigma_{S_{MK}}}, & S_{MK} > 0 \\ 0, & S_{MK} = 0. \\ \frac{S_{MK}+1}{\sigma_{S_{MK}}}, & S_{MK} < 0 \end{cases}$$

(8)

The null hypothesis is confirmed in a two-sided test if $|Z| \leq Z_{\alpha/2}$ at the $\alpha$-level of significance. A positive value of $Z_{MK}$ indicates an upward trend in the time series dataset; a negative value indicates a downward trend in the data.

*3.4. Sen's Slope Estimator*

While the Mann–Kendall test is a classical statistical test that identifies the trend component in a time series, it cannot quantify the magnitude or rate of the change (i.e., the trend's slope). Sen's slope estimator is a nonparametric procedure that can be used to estimate the slope of the linear trend (i.e., the linear rate of change per unit of time). Sen's slope estimator has been verified for its applicability to hydro-climatic time series data [45,51,52]. Moreover, the rate of change can be computed [30,45]:

$$\xi = \text{median}\left(\frac{x_j - x_i}{j - i}\right) \forall j < i$$

(9)

In which $\xi$ = the median of all computed linear slopes.

*3.5. Spearman's Rank Correlation*

Spearman's rank correlation is the nonparametric version of the Pearson product-moment correlation, which measures the strength and direction of the monotonic relationship between two ranked variables. Spearman's is a robust and resistant method [17,53]. The null hypothesis is that there is no statistically significant correlation between paired variables. The alternative hypothesis is that there is a monotonic relationship between the given pairs in the time series. This method has been used to investigate any correlations between paired hydro-climatic variables [17,54]. Spearman's rank correlation coefficient ranges between $-1$ and 1; negative correlations imply an inverse relationship between paired variables [55]. Accordingly, Spearman's rank correlation coefficient ($\vartheta$) is given by [53]:

$$\vartheta = 1 - \frac{6 \sum_{i=1}^{N} d_i^2}{N(N^2 - 1)}$$

(10)

where $d_i$ = the difference in the ranks of the values of the two given paired variables. The test statistic ($Z_S$) is calculated by [45]:

$$Z_S = \vartheta \sqrt{N - 1}.$$

(11)

If $|Z_S| > Z_\alpha$, the null hypothesis is rejected.

### 3.6. Artificial Neural Network (ANN)

ANN is a data-driven technique widely applied in time series forecasting and anomaly detection [56]. ANN is well known for its capability to deal with non-stationary data. Among many different kinds of neural networks, the multilayer perceptron (MLP) received the most attention due to its simplicity and capability to solve complex nonlinear problems [57]. As such, MLP is used in this paper to detect the change point of lake surface area and predict the future behavior of lake surface area.

The MLP is structured with a particular topology consisting of connected neurons and layers. Generic architectures of MLP networks are depicted in Figure 3. The input values are transmitted through links and neurons to the output values. All links between each neuron have an associated weight to scale the value traveling on that link [58]. During the model training process, the weight of each connection is adjusted iteratively to minimize a "cost function" such as the mean squared error (MSE). Once the topology of the MLP is designated, the model could learn the data behavior automatically and generate a mapping function to represent the mathematical relationship between inputs and outputs as:

$$Y_j = f\left(\theta_j + \sum_{i=1}^{n} w_{ji} X_i\right) \tag{12}$$

where $X_i$ = the ith input variable; $Y_j$ = the jth output variable; $\theta_j$ = the bias in the hidden layer; $n$ is the number of neurons in the hidden layer; $w_{ji}$ = the connection weight; and $f$ = the transfer function between layers.

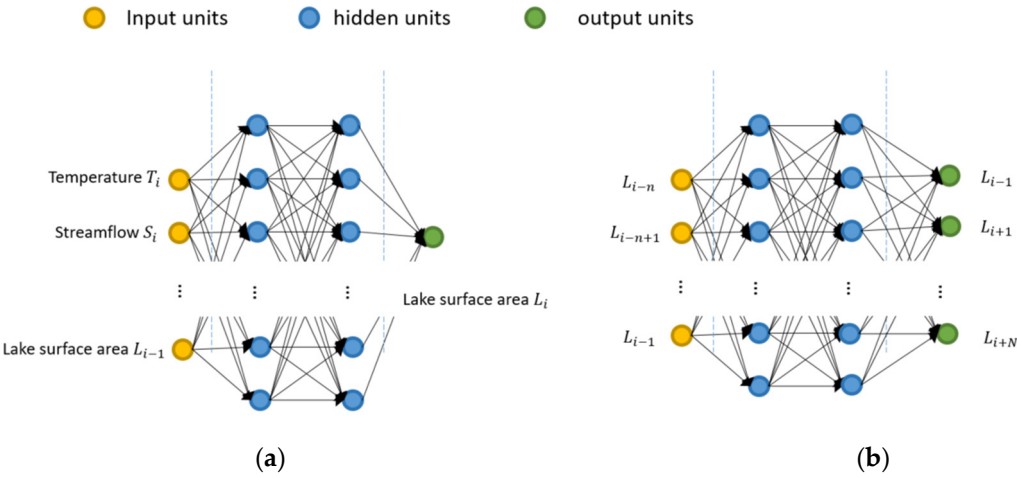

**Figure 3.** Overview of the topology of the two proposed MLP models. (**a**) is MLP model using extraneous factors and (**b**) MLP model using historical date of lake surface area.

The MLP model used in this paper is trained with backpropagation. The model is built using the Keras package in Tensorflow, all of which are coded in Python. Each layer is densely connected to the adjacent layers. The hyperparameters of the MLP model (e.g., the number of neurons or the number of layers, the learning rate) are determined based on the grid search strategy, and a detailed description of the parameter settings is summarized in Table 1. As shown in Figure 3, two MLP models are developed in this study. The first model (Figure 3a) is used to map the relationship between the extraneous factors (e.g., temperature, streamflow, rainfall) and the lake's surface area. The second model (Figure 3b) is used to generate a multi-step forecasting model based on the historical data of lake surface area to predict its future behavior. It should be noted that the dataset has been divided into two mutually exclusive sets, where 60% has been used as the training set and the remaining 40% has been used to test the trained model.

**Table 1.** Hyperparameters settings of the proposed ANN models.

| Parameter | Model I | Model II |
|---|---|---|
| Input | Rainfall, temperature, streamflow, lake's surface area from last time step | Last 24 steps of lake's surface area data |
| layers | 2 | 2 |
| No. of neurons of 1st layer | 25 | 20 |
| No. of neurons of 2nd layer | 10 | 15 |
| Output | One step of lake's surface area data | 12 steps of lake's surface area data |
| Activation function between hidden layers | Relu | Relu |
| Activation function between hidden layers and output layers | Linear | Linear |
| Learning rate | 0.01 | 0.01 |
| Optimizer | Adamax | Adamax |
| Loss function | Mean squared error | Mean squared error |

## 4. Results and Discussion

The monthly fluctuations in Maharlou Lake's surface over the period from 1992 to 2017 are depicted in Figure 4. While the lake's surface area shrank from May to October (i.e., as highlighted by lower average values in these months), the interquartile range (IQR) for these particular months is higher, which indicates that the lake's area exhibits more significant fluctuation during these months. The annual average of the lake's surface area was graphed from 1992 to 2017 (Figure 5). The Pettit test has identified a significant shift in the time series, which occurred in 2007. Statistically speaking, the said test indicates that the lake's surface area experienced such an abrupt change that the data series can be divided into pre-change-point and post-change-point sets.

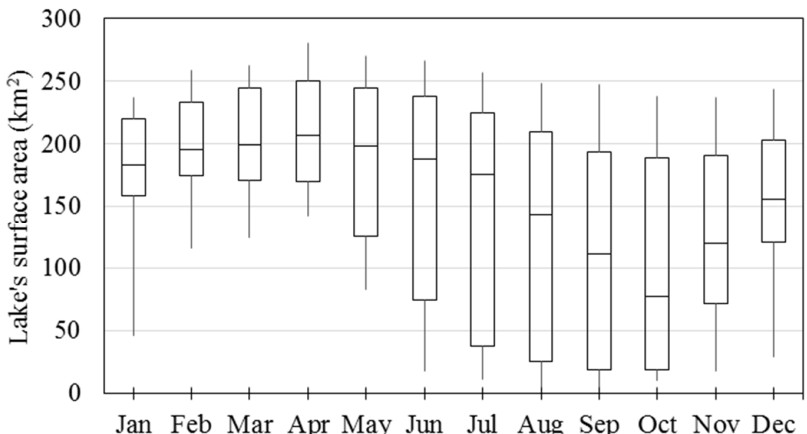

**Figure 4.** Average monthly variation of Maharlou Lake's surface area from 1992 to 2017.

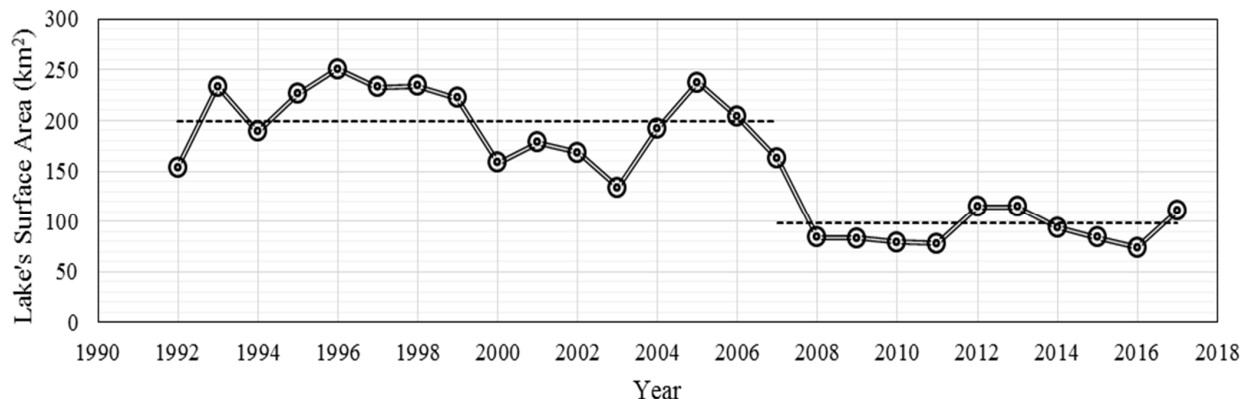

**Figure 5.** Annual average of Maharlou Lake's surface area.

Each month's time series was isolated to investigate this alleged abrupt change further. As such, each was represented by a series that showed the function of the lake's surface area in the said month from 1992 to 2017. Similarly, the Pettit test was employed to identify any potential jumping points in these series. The monthly changing points for each month were determined (Table 2). Apart from November and December, for which jumping points could be identified in 2001 and 2006, respectively, the rest of the months experienced breaking points in 2007.

**Table 2.** The results of the detected changing points in the lake's surface area.

|  | **Jan** | **Feb** | **Mar** | **Apr** | **May** | **Jun** | **Jul** | **Aug** | **Sep** | **Oct** | **Nov** | **Dec** |
|---|---|---|---|---|---|---|---|---|---|---|---|---|
| Jumping point | 2007 | 2007 | 2007 | 2007 | 2007 | 2007 | 2007 | 2007 | 2007 | 2007 | 2001 | 2006 |

As stated earlier, one can also resort to data mining methods to provide additional context about the hidden structures in the dataset. On that note, an MLP model (model I) was trained to simulate the lake's surface area time series using extraneous factors. Figure 6 visualizes the overall performance of the calibrated model I. Figure 6a compares the obtained results from the calibrated model I against the observed data, and Figure 6b shows the residual values that represent the absolute prediction error at each timestep. Based on the obtained results, model I's performance can be broken down into two sections, where the first portion represents the model's results before 2008 and the second section denotes the 2008-onward data. As can be seen in Figure 6, the accuracy of model I in the first portion of the dataset can be deemed acceptable. However, the prediction accuracy of the regression model drastically declines in the second portion of the simulation. This, in and of itself, hints that the underlying structure of the data has experienced a sudden change, to the point that the calibrated mode cannot accurately capture the behavior of the second portion of the time series.

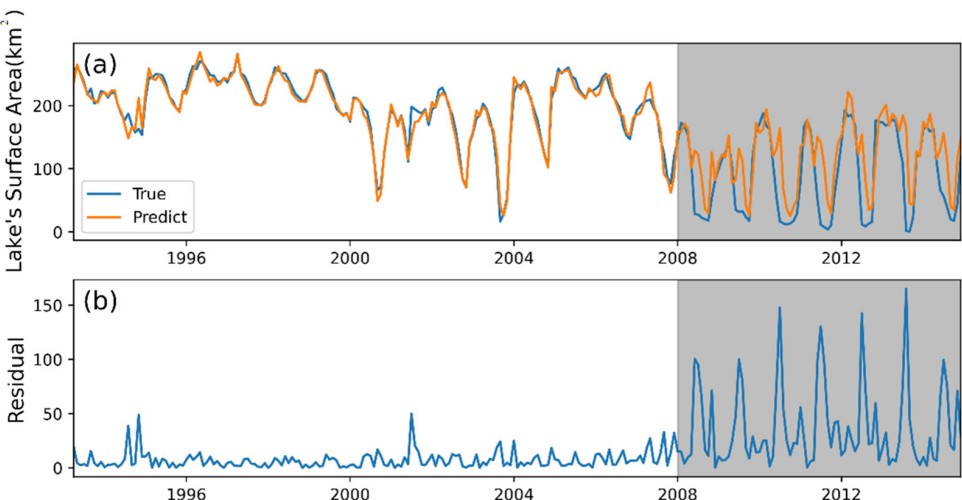

**Figure 6.** Prediction results of the lake's surface area obtained from MLP model I. (**a**) Predictions for the lake's surface area (km$^2$) and (**b**) Residual from the model.

The next stage of the investigation is the identification of any potential gradual changes in the structure of the dataset using statistical analysis. The Mann–Kendall test and Sen's slope estimator were used to identify the gradual changes in surface area (Table 3). The results show a statistically significant negative trend component in each of the lake's surface area time series. These components are more pronounced from June to October; moderate in May, November, and December; and mild from January to April. From a statistical point of view, the results indicate that the lake's surface area is showing both abrupt and gradual changes, which can be confirmed by the change-detection analysis of Maharlou Lake (Figure 7).

**Table 3.** Detected trend component in the lake's surface area time series.

|  | Jan | Feb | Mar | Apr | May | Jun | Jul | Aug | Sep | Oct | Nov | Dec |
|---|---|---|---|---|---|---|---|---|---|---|---|---|
| Trend component | −3.98 | −3.31 | −4.00 | −3.84 | −5.45 | −7.47 | −8.71 | −8.99 | −8.69 | −8.11 | −5.66 | −5.09 |

Note: These values were deemed statistically significant at $\alpha = 0.05$.

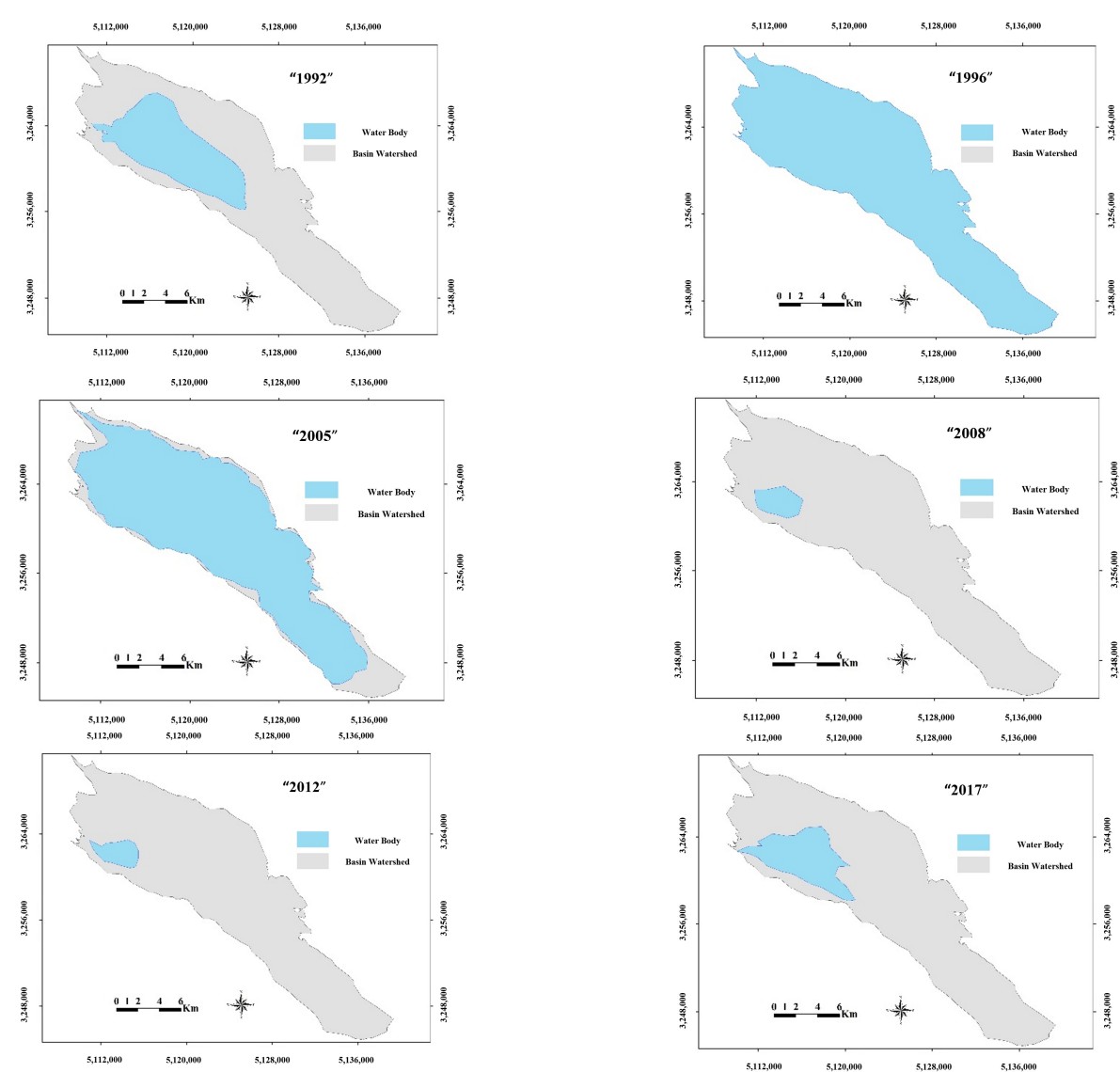

**Figure 7.** Change detection of Maharlou Lake's surface.

In addition to the above analysis, it was also essential to analyze the components of the hydro-climatic time series and, in turn, identify any anomalies in the behaviors of the hydro-climatic variables (e.g., rainfall, temperature, and streamflow) during the study period. An aberration in the natural fluctuations of these variables could signify the presence of external stressors that could change or have changed the *status quo* of Maharlou Lake. Therefore, a series of statistical tests were used to detect irregularities in the behaviors of the variables. Tables 4–6 summarize the detected changing points among the rainfall, temperature, and streamflow datasets that were deemed statistically significant. While the detected change points occurred from the late 1980s to the mid-2000s, most of the jumps occurred in 1998 and 1999. The temperature data exhibit the highest number of statistically significant jump points, while the rainfall data contain the least number of jump points (all of these occur in March). All of the monitoring stations in the vicinity of the lake (including Dobaneh and Shiraz synoptic stations) reveal that change-points occurred in 1999. The

annual time series of all hydro-climatic variables are homogeneous, and the change-points in these data were not found to be statistically significant.

**Table 4.** Changing points detected in the monthly rainfall time series of selected stations.

|  | Jan | Feb | Mar | Apr | May | Jun | Jul | Aug | Sep | Oct | Nov | Dec |
|---|---|---|---|---|---|---|---|---|---|---|---|---|
| Dobaneh | - | - | 1999 | - | - | - | - | - | - | - | - | - |
| Ghalat | - | - | 1999 | - | - | - | - | - | - | - | - | - |
| Mehrabad-Ramjerd | - | - | 1999 | - | - | - | - | - | - | - | - | - |
| Sarvestan | - | - | 1999 | - | - | - | - | - | - | - | - | - |
| Shiraz (Sazman-e-Ab) | - | - | 1999 | - | - | - | - | - | - | - | - | - |

Note: These values were deemed statistically significant at $\alpha$ = 0.05.

**Table 5.** Changing points detected in the monthly temperature time series of selected stations.

|  | Jan | Feb | Mar | Apr | May | Jun | Jul | Aug | Sep | Oct | Nov | Dec |
|---|---|---|---|---|---|---|---|---|---|---|---|---|
| Fasa | - | - | 1999 | 1998 | 1998 | 1997 | 2002 | 2000 | 2009 | 2000 | - | - |
| Sad-e-Dorodzan | - | - | 1999 | - | - | 2005 | 2005 | - | - | 1998 | - | - |
| Shiraz (Synoptic) | - | 1998 | 1999 | - | - | 1997 | - | - | - | 1996 | - | - |
| Zarghan | - | - | 1999 | - | 1998 | 2005 | 2005 | - | - | 2000 | - | - |

Note: These values were deemed statistically significant at $\alpha$ = 0.05.

**Table 6.** Changing points detected in the monthly streamflow time series of selected stations.

|  | Jan | Feb | Mar | Apr | May | Jun | Jul | Aug | Sep | Oct | Nov | Dec |
|---|---|---|---|---|---|---|---|---|---|---|---|---|
| Chenar | - | - | - | - | - | - | - | - | - | - | - | 1987 |
| Chenar-Sokhteh | - | - | - | - | 1999 | - | - | - | - | - | - | - |
| Pol-e-Fasa | 2006 | 2006 | 1999 | 1999 | 1999 | - | - | - | - | - | - | - |

Note: These values were deemed statistically significant at $\alpha$ = 0.05.

The Mann–Kendall test and Sen's slope estimator were used to identify and measure the gradual changes in patterns in the hydro-climatic data (Tables 7–9). Rainfall shows the most change per unit of time. In contrast, the temperature data displayed the least amount of change, though the gradual temperature change was an upward trend. In the case of streamflow, the data showed only downward trends. The rainfall data displayed a negative trend in March, but an upward trend appeared in November.

**Table 7.** Detected trend component in the monthly rainfall time series of selected stations.

|  | Jan | Feb | Mar | Apr | May | Jun | Jul | Aug | Sep | Oct | Nov | Dec |
|---|---|---|---|---|---|---|---|---|---|---|---|---|
| Dobaneh | - | - | −1.75 | - | - | - | - | - | - | - | - | - |
| Ghalat | - | - | −2.33 | - | - | - | - | - | - | - | 1.44 | - |
| Mehrabad-Ramjerd | - | - | - | - | - | - | - | - | - | - | - | - |
| Sarvestan | - | - | −1.40 | - | - | - | - | - | - | - | 0.20 | - |
| Shiraz (Sazman-e-Ab) | - | - | −1.76 | - | - | - | - | - | - | - | 0.53 | - |

Note: The trends that were significant at $\alpha$ = 0.05 were measured by Sen's slope test and reported here.

**Table 8.** Detected trend component in the monthly temperature time series of selected stations.

|  | Jan | Feb | Mar | Apr | May | Jun | Jul | Aug | Sep | Oct | Nov | Dec |
|---|---|---|---|---|---|---|---|---|---|---|---|---|
| Fasa | - | 0.07 | 0.10 | 0.07 | 0.09 | 0.08 | 0.07 | 0.04 | 0.04 | 0.06 | - | - |
| Sad-e-Dorodzan | - | 0.08 | - | - | 0.05 | 0.08 | 0.05 | - | - | 0.06 | - | - |
| Shiraz (Synoptic) | - | 0.07 | 0.09 | - | - | 0.04 | - | - | - | - | - | - |
| Zarghan | - | 0.05 | 0.08 | - | 0.07 | 0.05 | - | - | - | 0.05 | - | - |

Note: The trends that were significant at $\alpha$ = 0.05 were measured by Sen's slope test and reported here.

**Table 9.** Detected trend component in the monthly streamflow time series of selected stations.

| | Jan | Feb | Mar | Apr | May | Jun | Jul | Aug | Sep | Oct | Nov | Dec |
|---|---|---|---|---|---|---|---|---|---|---|---|---|
| Chenar | - | - | - | - | - | - | - | - | - | - | - | - |
| Chenar-Sookhteh | - | - | - | - | −0.01 | - | - | - | - | - | - | - |
| Pol-e-Fasa | −0.11 | −0.18 | −0.23 | −0.10 | −0.05 | - | - | - | - | - | - | −0.07 |

Note: The trends that were significant at $\alpha = 0.05$ were measured by Sen's slope test and reported here.

While detecting gradual and abrupt changes in the hydro-climatic variables could be interpreted as signals for climate change and anthropogenic drought, respectively, linking these changes to those identified by monitoring the lake's surface area requires further analysis. From a statistical analysis standpoint, one can attempt to relate the abrupt changes in surface area to those processes revealed by the time series of hydro-climatic variables. To unravel the potential cause behind the withering of the lake, Spearman's test was used to identify the statistically significant correlations between surface area and hydro-climatic variables (Table 10). The results show that apart from August and December, the surface area during the rest of the months is statistically related to the hydro-climatic variables; these correlations are positive for rainfall and streamflow (i.e., the increase in rain and flow correlates to increasing surface area) and negative for temperature (i.e., increasing temperature correlates with decreasing surface area). The obtained results from the correlation analysis are also in line with the findings from the data mining methods. The point being that the trained MLP model (model I) was able to accurately emulate the patterns in the lake's surface area data set through the extraneous factors before the occurrence of the abrupt change in the structure of the data. Thus, it is safe to assume that the changes in the hydro-climatic variables would impact the lake's area. As for the reason the shrinkage is more pronounced in June to October, one could recall the hydro-climatic conditions in these particular months (Figure 2). During those months, rainfall and streamflow are virtually nonexistent, and the temperature has a more significant effect, evaporating the lake's water. The lake's surface area (Figure 4) and volume are at their lowest points at that time. As such, the same amount of water loss that may be happening during periods of greater input (i.e., rainfall) would result in a more pronounced depletion rate.

**Table 10.** Spearman's test results to identify any correlation between lake's surface area and hydro-climatic variables.

| | Temperature | Rainfall | Streamflow |
|---|---|---|---|
| Jan | | | 0.672 |
| Feb | | | 0.718 |
| Mar | −0.614 | 0.484 | 0.758 |
| Aar | | | 0.771 |
| May | −0.624 | | 0.759 |
| Jun | −0.619 | 0.525 | 0.708 |
| Jul | −0.482 | 0.292 | |
| Aug | | | |
| Sea | −0.447 | | |
| Oct | −0.442 | | |
| Nov | | | 0.420 |
| Dec | | | |

Note: Only the correlation coefficients that were significant at $\alpha = 0.05$ were reported here.

As the final phase of this investigation, a data mining-based model (model II) was used to understand the current status of the lake. Figure 8 shows the predicted behavior of Maharlou Lake's surface area in the future based on the MLP model II mentioned in Section 3.6. The obtained results from the MLP model (model II) show that the emulated patterns in the lake's surface area dataset closely resemble the same behavior observed

after the identified abrupt change in 2007/2008. This goes to show that the lake has undergone a morphological change since the change point and is currently adopting this new equilibrium.

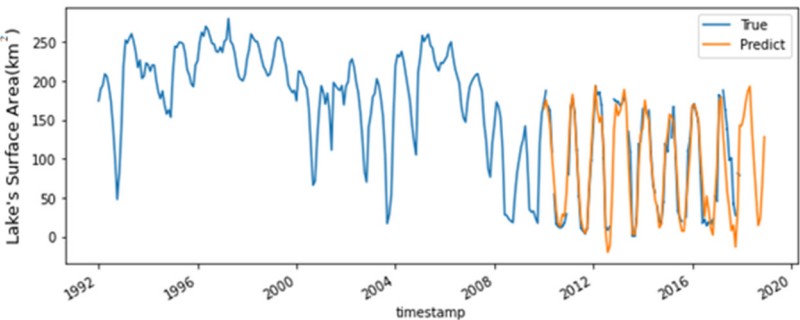

**Figure 8.** Prediction results of lake's surface area based on MLP model II.

It should be noted, however, that the depletion patterns in Maharlou Lake are slightly different from the patterns at Urmia Lake, another iconic inland water body in Iran. Though the early signs for both occurred in the same period (i.e., the late 1990s), the changing patterns in the climatic variables (i.e., precipitation and temperature) were not as pronounced in Maharlou's basin as they were in Urmia's basin [17,59]. In other words, Urmia's deterioration was due to direct anthropogenic stress that changed the region's water budget. As for the Maharlou, while one cannot dismiss direct adverse impacts from people's activities, the changes in the hydro-climatic variables also played a crucial role in the deterioration of the lake. This makes the rectification of this case a bit more challenging. It is worth noting that further studies are needed to reveal the precise impact of direct human impact on the lake's shrinking pattern.

## 5. Conclusions

Whether it is the increasing human population or the impacts of climate change, sustainable management of water resources is proving to be more challenging than ever before. One of the frontiers of this continuous struggle is the preservation of inland water bodies such as saline lakes. Understanding the nature of lakes' depletion could help decision-makers cope with the problem by implementing a lake recovery and protection strategy, mitigating the adverse impacts of the loss of the lakes, and/or adapting to the new situation. This study presents a comprehensive analytical framework that can reveal depletion patterns using RS, statistical analysis, and data mining techniques. Maharlou Lake, Iran, was used as a case study to demonstrate the application of the proposed framework.

The results reveal specific depletion patterns in Maharlou's surface area. The shrinking pattern of Maharlou Lake occurred in two different ways: a sharp reduction in 2007/2008 and a gradual decline that was more pronounced during dry seasons and less noticeable during wet seasons. Hydro-climatic variables (i.e., rainfall, temperature, and streamflow) were tested for any statistically significant patterns of change. This revealed that there were both abrupt and gradual changes in these variables' time series, and the changes could be signaling the impacts of climate change and anthropogenic drought. The analysis of the relationships between the hydro-climatic patterns and the surface area patterns indicated that the correlations were complicated. The magnitude of the lake's depletion was determined in part by the season. For instance, when the lake's storage was low in the dry season, the recharge components (i.e., rainfall and streamflow) were virtually nonexistent, and temperatures were at their highest. During this time, depletion is more pronounced. The obtained results from data mining models show that Maharlou Lake is undergoing a permanent morphological change. If preventive measures are not taken soon, the lake could potentially pass a point of no return, after which recovery of the lake and preservation of the associated ecosystems and dependent economic activities would

be next to impossible. Preventive measures such as incorporating long-term watershed strategies may provide a remedy to mitigate these observed patterns.

It should be noted, however, that the temporal depletion patterns of Maharlou Lake are somewhat different from the temporal depletion patterns of Urmia Lake, a similar saline lake in Iran. Urmia's deterioration arose primarily from anthropogenic causes that upset the region's water budget. While one cannot dismiss the direct adverse impact of human activities, the changes in the hydro-climatic variables in the Maharlou Basin have also led to the deterioration of the lake. The complex nature of the causes makes the management of this case more challenging.

**Author Contributions:** All authors contributed to the study conception and design. Material preparation, data collection, and analysis were performed by B.Z.-A., M.N.J. (Mojtaba Naghdyzadegan Jahromi), X.W., M.E. and M.N.J. (Maryam Naghdizadegan Jahromi) The first draft of the manuscript was written by B.Z.-A. and all authors commented on previous versions of the manuscript. All authors have read and agreed to the published version of the manuscript.

**Funding:** The authors thank the University of Exeter for providing the required funding to publish this open access paper.

**Data Availability Statement:** The authors declare all data and materials, as well as software applications or custom codes, are in line with published claims and comply with field standards. Furthermore, the research data supporting this publication are provided within this paper.

**Conflicts of Interest:** The authors declare no conflict of interest.

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
