# Peer review of "Uncovering the Depletion Patterns of Inland Water Bodies via Remote Sensing, Data Mining, and Statistical Analysis"

_water, doi:10.3390/w15081508_

Round 1
Reviewer 1 Report
The article addresses the interesting issue of lake decay. The paper uses standard methods for this type of analysis (Pettit test, MK test, Spearman test). I believe that the article has an applied character, but before publication it requires consideration of the following comments:
Figure 1. The first figure is redundant-it should be included as a thumbnail within the figure below. In more detail, please approximate the catchment use structure, terrain elevation within the catchments and hydrographic network.
Among the hydroclimatic variables, please include the amount of evaporation. This is a key element of the water balance- especially for the climatic zone under study.
What is the human impact on water level fluctuations, which is an important component of this type of oscillation. Does such activity exist in the analyzed catchment and if so, to what extent? At this point nothing is known about it. In addition, the article contains statements:
„As for the Maharlou, while one can-398 not dismiss direct adverse impacts from people’s activities, the changes in the hydro-cli-399 matic variables also played a crucial role in the deterioration of the lake”.(line 398-400)
„While one cannot dismiss the direct adverse impact of human activities, the changes in the hydro-climatic variables in the Maharlou basin have also led to the deterioration of the lake”. (Line 432-434).
Thus, the authors are aware of the role of this factor and the research task is to clearly confirm or exclude this type of activity.
Author Response
Reviewer #1:
The article addresses the interesting issue of lake decay. The paper uses standard methods for this type of analysis (Pettit test, MK test, Spearman test). I believe that the article has an applied character, but before publication it requires consideration of the following comments:
Reply:
We would like to thank the anonymous reviewer for these valuable suggestions. We did our best to address and reflect on all these points in the revised paper. It is worth noting that the paper has been reviewed and approved by all authors and native English speakers to ensure the final quality of the work.
Figure 1. The first figure is redundant-it should be included as a thumbnail within the figure below. In more detail, please approximate the catchment use structure, terrain elevation within the catchments and hydrographic network.
Reply:
Per the suggestion of the reviewers the said figure has been revised accordingly.
Among the hydroclimatic variables, please include the amount of evaporation. This is a key element of the water balance- especially for the climatic zone under study.
Reply:
While this is indeed a valid and insightful point, the reason we were not able to initially include this variable in our study was twofold; First and most significant problem was the lack of reliable data in the long period required for both statistical tests and machine learning models. The other, however, that helped reduce the urgency of using this data, at least in a direct way, was the known direct strong correlation between evapotranspiration and temperature. Established and verified models such as Penman–Monteith or Hargreaves-Samani equations showcase how you can calculate evapotranspiration from temperature, granted that you often need additional information such as humidity, which is often related to precipitation and temperature, both of which have been explored in this study. Due to both these reasons (i.e., lack of reliable long-term datasets and the strong relationship between the said variable and other explored variables studies in this paper), we were forced to only rely on the selected variables.
What is the human impact on water level fluctuations, which is an important component of this type of oscillation. Does such activity exist in the analyzed catchment and if so, to what extent? At this point nothing is known about it. In addition, the article contains statements:
“As for the Maharlou, while one cannot dismiss direct adverse impacts from people’s activities, the changes in the hydro-climatic variables also played a crucial role in the deterioration of the lake”.(line 398-400)
“While one cannot dismiss the direct adverse impact of human activities, the changes in the hydro-climatic variables in the Maharlou basin have also led to the deterioration of the lake”. (Line 432-434).
Thus, the authors are aware of the role of this factor and the research task is to clearly confirm or exclude this type of activity.
Reply:
As you keenly mentioned, we have stated these to show that while such impacts, though not clear at this stage, can be influential in the lake’s behavior, we proved that the observed patterns are certainly beyond this potential influence. These are interesting research questions that need to be studied in future studies, and the paper was revised accordingly to shed light on this crucial matter.
At the end, we want to extend our utmost gratitude to the anonymous reviewer for these constructive and insightful comments, which did indeed help improve the quality of our work.
Reviewer 2 Report
Figure 1: weather stations were not marked.
Eq (2): define t.
The review suggests authors to briefly investigate the impact of groundwater depletion due to urbanization on the volume of water body
Lines 425-428: Authors’ suggestion is a bit vague and weak. It is recommended to list possible preventive practices considering the watershed conditions and long-term watershed planning.
Author Response
Reviewer #2:
Figure 1: weather stations were not marked.
Reply:
Per the suggestion of the review, the said figure has been revised accordingly.
Eq (2): define t.
Reply:
That merely is a representation of a selected time step. The paper was revised accordingly to reflect this.
The review suggests authors to briefly investigate the impact of groundwater depletion due to urbanization on the volume of water body.
Reply:
It is our understanding that there is not notable charging point from the region’s groundwater system. The article was revised to reflect this idea.
Lines 425-428: Authors’ suggestion is a bit vague and weak. It is recommended to list possible preventive practices considering the watershed conditions and long-term watershed planning.
Reply:
This was indeed a quite valuable suggestion. We revised the paper accordingly to reflect this idea.
At the end, we want to extend our utmost gratitude to the anonymous reviewer for these constructive and insightful comments, which did indeed help improve the quality of our work.
Reviewer 3 Report
Addressing the issue of shrinking saline lakes around the globe has turned into one of the most pressing issues for sustainable water resources management. While it has been established that natural climate variability, human interference, climate change, or a combination of these factors can lead to the depletion of saline lakes, it is crucial to investigate each case and diagnose the potential causes for this devastating phenomenon. This study aims to promote a comprehensive analytical framework that can reveal any significant depletion patterns in lakes while analyzing the potential reasons behind these observed changes. The topic is interesting and within the scope of Water. The main conclusions are supported by results. I would recommend a Minor Revision.
(1) This manuscript uncovered the depletion patterns of inland water bodies via remote sensing, data mining, and statistical analysis; however, the quantitative results are missing in Abstract and Conclusions.
(2) Introduction: The inland water bodies are impacted by natural climate variability, human interference and anthropogenic climate change. The present Introduction is not sufficient. I would recommend providing a more comprehensive review about climate change impacts on water cycle. The following references may be helpful.
https://doi.org/10.1038/s41612-022-00235-9
https://doi.org/10.1038/s41893-022-01024-1
https://doi.org/10.1038/s41558-021-01044-3
(3) The figure quality is low; please improve the figure rendering.
(4) The uncertainty should be briefly discussed.
(5) The underlying physical mechanism should be discussed.
Author Response
Reviewer #3:
(1) This manuscript uncovered the depletion patterns of inland water bodies via remote sensing, data mining, and statistical analysis; however, the quantitative results are missing in Abstract and Conclusions.
Reply:
While this is often a valid point, and for that, we thank the reviewer for bringing this up, it is, we believe, not quite applicable to our case. The point is that the result of the numeric analysis conducted in this study is, for the most part, relative and, as such, would not convey any meaningful information in and of itself. The point is that, for instance, the result of statistical results, and so on and so forth, is just to unveil a depletion pattern in the lake, which we have covered in both the Abstract and the Conclusions. We have even gone so far as to mention the year in which, for instance, a jumping point was observed in the data.
(2) Introduction: The inland water bodies are impacted by natural climate variability, human interference and anthropogenic climate change. The present Introduction is not sufficient. I would recommend providing a more comprehensive review about climate change impacts on water cycle. The following references may be helpful.
https://doi.org/10.1038/s41612-022-00235-9
https://doi.org/10.1038/s41893-022-01024-1
https://doi.org/10.1038/s41558-021-01044-3
Reply:
These are pretty relevant and interesting papers that could help provide the readers with a much richer literature review and were used to update the introduction section. It is worth noting that the raised point was also mentioned explicitly in the introduction section, and as such, these papers were used to emphasize this argument.
(3) The figure quality is low; please improve the figure rendering.
Reply:
All the figures were checked to ensure that they have the minimum quality standards required by the journal.
(4) The uncertainty should be briefly discussed.
Reply:
This is quite a valuable point and one of this study's primary themes, which is why we explored the data using both statistical tests and machine learning models. That said, we have revised the paper, most notably the introduction section, to better shed light on this matter.
(5) The underlying physical mechanism should be discussed.
Reply:
While this is a valuable point, in and of itself, the central theme of this paper, as discussed extensively in the instruction section, was to find an alternative and reliable solution to circumvent the challenges of simulating or unraveling the actual physical mechanisms. The idea is that this phenomenon is often quite complex, and there are no definitive solutions to explain them, again, as supported by the papers mentioned in the instruction section. The idea here, however, was to use some computational intelligence-based methods, such as Machine learning models, and using Remote sensing data, coupled with traditional statistical tests to explore this matter from a different angle in a reliable way. All that said, the paper was revised to better reflect on these nuances and shed light on the importance of this framework.
At the end, we want to extend our utmost gratitude to the anonymous reviewer for these constructive and insightful comments, which did indeed help improve the quality of our work.